# Machine learning models to accelerate the design of polymeric long-acting injectables

Pauric Bannigan[1], Zeqing Bao[1], Riley J. Hickman[2,3,4], Matteo Aldeghi[2,3,4], Florian Häse [2,3,4], Alán Aspuru-Guzik [2,3,4,5,6,7,8] ✉ & Christine Allen [1,8] ✉

Long-acting injectables are considered one of the most promising therapeutic strategies for the treatment of chronic diseases as they can afford improved therapeutic efficacy, safety, and patient compliance. The use of polymer materials in such a drug formulation strategy can offer unparalleled diversity owing to the ability to synthesize materials with a wide range of properties. However, the interplay between multiple parameters, including the physico-chemical properties of the drug and polymer, make it very difficult to intuitively predict the performance of these systems. This necessitates the development and characterization of a wide array of formulation candidates through extensive and time-consuming in vitro experimentation. Machine learning is enabling leap-step advances in a number of fields including drug discovery and materials science. The current study takes a critical step towards data-driven drug formulation development with an emphasis on long-acting injectables. Here we show that machine learning algorithms can be used to predict experimental drug release from these advanced drug delivery systems. We also demonstrate that these trained models can be used to guide the design of new long acting injectables. The implementation of the described data-driven approach has the potential to reduce the time and cost associated with drug formulation development.

Long-acting injectables (LAI) are a class of advanced drug delivery systems that are designed to release their cargo over extended periods of time in order to achieve a prolonged therapeutic effect. LAIs that are amenable to parenteral administration can confer several advantages over conventional drug formulations, including increased patient compliance and bioavailability of drug[1]. Moreover, LAIs can be engineered to provide either local (e.g., Zilretta®) or systemic (e.g., Lupron Depot®) drug exposure over a prolonged period, making them ideal formulation strategies for the treatment of chronic diseases[2]. The unparalleled chemical and physical diversity afforded by these materials makes polymer-based LAIs a particularly apt version of this drug delivery strategy. These systems can be engineered to entrap drugs within a polymer matrix with release occurring via various mechanisms, including erosion, diffusion, or simultaneous erosion and diffusion[3]. In addition to achieving sustained or controlled drug release, the encapsulation of drugs into these polymeric matrices can often protect therapeutic cargo[4]. To date, various polymer-based LAI technologies administered via the intramuscular[5], subcutaneous[6], and intra-articular[7] routes have received regulatory approval (Fig. 1a).

[1]Leslie Dan Faculty of Pharmacy, University of Toronto, Toronto, ON M5S 3M2, Canada. [2]Department of Computer Science, University of Toronto, Toronto, ON M5S 3H6, Canada. [3]Department of Chemical Engineering & Applied Chemistry, University of Toronto, Toronto, ON M5S 3E5, Canada. [4]Vector Institute for Artificial Intelligence, Toronto, ON M5S 1M1, Canada. [5]Department of Materials Science & Engineering, University of Toronto, Toronto, ON M5S 3E4, Canada. [6]Lebovic Fellow, Canadian Institute for Advanced Research, Toronto, ON M5S 1M1, Canada. [7]CIFAR Artificial Intelligence Research Chair, Vector Institute, Toronto, ON M5S 1M1, Canada. [8]These authors jointly supervised this work: Alán Aspuru-Guzik, Christine Allen. ✉e-mail: alan@aspuru.com; cj.allen@utoronto.ca

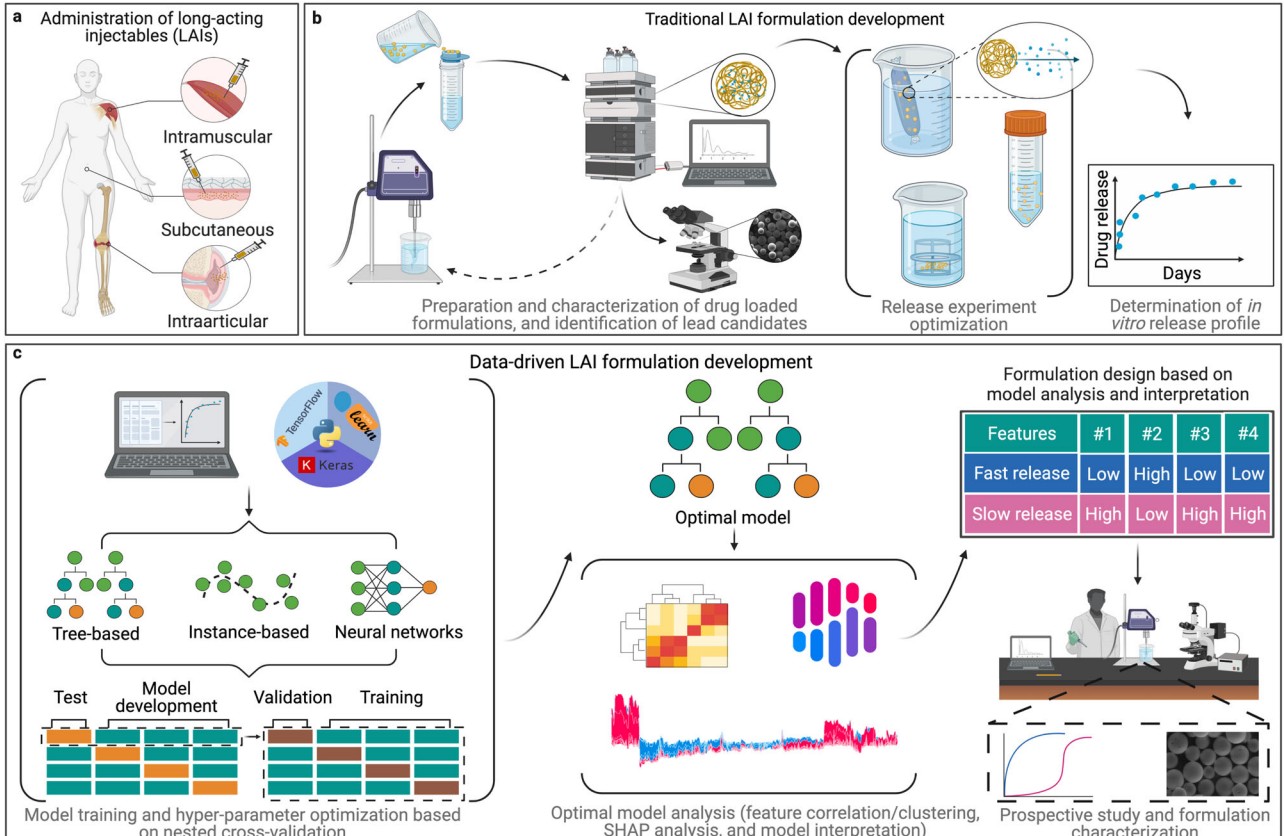

**Fig. 1 | Schematic demonstrating traditional and data-driven formulation development approaches for long-acting injectables (LAIs). a** Selected routes of administration for FDA-approved LAI formulations. **b** Typical trial-and-error loop commonly employed during the development of LAIs termed "traditional LAI formulation development". **c** Workflow employed in this study to train and analyze machine learning (ML) models to accelerate the design of new LAI systems, termed "Data-driven LAI formulation development".

Despite the advantages associated with polymeric LAIs, their translation from bench to bedside remains non-trivial. In the past two decades, only about 30 polymeric LAI products have received regulatory approval, in contrast to the thousands of conventional oral formulations approved in the same period[3,8]. Several pivotal challenges limit the development and clinical translation of polymeric LAIs. Firstly, there are few biodegradable polymer materials that are generally recognized as safe (GRAS) for parenteral administration. To date, the polymeric LAIs that have received clinical approval are largely based on a single polymer: poly(lactide-co-glycolide) (PLGA)[2,3,9]. The use of a material, such as PLGA, with an established safety profile, may accelerate the regulatory approval of new LAI formulations. However, it is well recognized that drug-polymer compatibility significantly influences the performance of a formulation, including drug loading capacity, drug release, and stability[10]. Given that each drug has its own unique physicochemical properties, it is unlikely that any one polymer material is ideally suited for the formulation of all drugs. Thus, reliance on the relatively small subset of polymeric materials that have GRAS status likely restricts our ability to develop polymer-based LAIs for many classes of drugs. Moreover, for a given polymer material, there is a wide range of variables that must be optimized during LAI preparation. Furthermore, changes to any of these variables have the potential to impact formulation performance, and the net effect of such alterations cannot be known a priori. Often, initially, promising formulations can fail at various stages during development due to unwanted drug release rates, making their re-formulation and re-evaluation necessary (Fig. 1b). This trial-and-error-based approach represents a significant bottleneck in LAI development[11].

To date, several strategies have been investigated to inform decision-making and expedite the drug formulation development process. For instance, mathematical models have been used to describe and greatly enhance our understanding of drug release mechanisms[12]. However, the application of these empirical models is limited to *post hoc* analysis of the in vitro drug release profiles of LAIs, and they do not offer information on in vitro drug release from LAIs a priori. More recently, molecular dynamics simulations have been investigated[13]. These techniques have been useful in quantifying links between drug release rates and formulation parameters (including particle size and drug loading levels)[14]. While the development of these techniques is an active area of research, molecular dynamics simulations of entire drug delivery systems are computationally intensive. These approaches can be used to confer useful information on potential LAI systems; however, they cannot currently be used in place of experimental drug release assays[14]. Several studies have also investigated machine learning (ML) approaches[11].

The application of ML in the pharmaceutical sciences is generally limited by a lack of available open-source datasets to train models[11]. Past efforts to predict in vitro drug release from LAIs using ML have exclusively considered neural network (NN) based models, and have examined narrow application domains. For example, Szlęk et al. used NNs to predict the drug release of proteins and peptides from PLGA-based microparticles (MPs) using a dataset of 68 PLGA formulations extracted from the literature[15]. Small molecule studies have also employed NNs to predict drug release, and these have generally been limited to less than 20 data instances[16]. The use of NNs for supervised learning tasks in the low-data regime may be associated with an increased risk of overfitting compared to alternative ML algorithms,

such as tree-based models or Gaussian processes, which are usually better suited for sparse data problems.

The current study pursued the investigation and refinement of ML models for accurate prediction of fractional drug release from polymeric LAIs (Fig. 1c). To achieve this, we trained and evaluated a series of eleven different ML algorithms, including multiple linear regression (MLR), MLR with least absolute shrinkage and selection operator regularization (lasso), partial least squares (PLS), decision tree (DT), random forest (RF), light gradient boosting machine (LGBM), extreme gradient boosting (XGB), natural gradient boosting (NGB), support vector regressor (SVR), k-nearest neighbors (k-NN), and NN. The LGBM model was found to have the best performance: predicting fractional drug release with a high degree of accuracy.

## Results and discussion
### Model selection
The dataset used to train ML models was constructed from previously published studies by our group and other research groups[10,17–44]. The studies performed by our group included spherical and cylinder-shaped polymeric LAIs[10,18,38]. Data from external sources were identified using the Web of Science search engine and the keyword combination "polymeric microparticle" and "drug delivery"[17,19–37,39–44]. In each research article selected for dataset construction, the in vitro release of the drug from the respective formulation was characterized. The final dataset was composed of descriptors for a range of small molecule drugs, polymer materials, and LAIs, as well as the in vitro drug release profiles and experimental conditions under which the drug release profiles were generated. In total, this included 181 drug release profiles with 3783 individual fractional release measurements for 43 unique drug-polymer combinations. The LAIs were mostly formed from commercially available polymers, such as PLGA, polylactic acid (PLA), and polycaprolactone (PCL), of various molecular weights and lactide-to-glycolide ratios (for PLGA systems). A visual representation of the fractional drug release profiles for all formulations within the collected dataset is shown in Supplementary Fig. 1. A summary of this dataset in terms of drug–polymer combinations is included in Supplementary Fig. 2.

Seventeen molecular and physicochemical descriptors were initially selected based on domain knowledge as input features to describe the LAI formulations of the various ML models. This included input features that described the physicochemical properties of the drug, polymer and LAI system and features that accounted for experimental conditions under which the in vitro release studies were conducted. As shown in Supplementary Table 1, the models were trained in such a way that for each drug release profile for a specific LAI, only the input feature, the timepoint for drug release measurements (*Time*), varied with all other input features remaining constant. These included the drug's molecular weight (*Drug_MW*), topical polar surface area (*Drug_TSPA*), number of heteroatoms (*Drug_NHA*), melting temperature (*Drug_Tm*), acid dissociation constant (*Drug_pKa*), and calculated partition coefficient (*Drug_LogP*) as well as polymer molecular weight (*Polymer_MW*), molar cross-linking ratio (*CL_Ratio*), and lactide-to-glycolide ratio (*LA/GA*), the drug loading capacity of the LAI expressed as a fraction (*DLC*), the initial drug-to-polymer ratio used to prepare the LAIs (*Initial D/M ratio*), the surface area-to-volume ratio of the LAI (*SA-V*), the percent of surfactant in the experimental release media (*SE*), fractional drug release at 6 h ($T = 0.25$), fractional drug release at 12 h ($T = 0.5$), and fractional drug release at 24 h ($T = 1.0$).

The selected panel of ML models was trained and evaluated using a nested cross-validation strategy that included an inner loop (i.e., model training and hyperparameter tuning) and an outer loop (i.e., model evaluation). Briefly, for each ML model, 20% of the drug–polymer groups in the dataset were randomly selected as a test set. The remaining 80% were used for model development. Within the inner loop, each model was subject to a hyperparameter optimization procedure using group k-fold ($k = 10$) cross-validation. Model hyperparameters were tuned using a random grid search, where the objective function was the average model performance across these k-fold groups of drug-polymer combinations. The range of model hyperparameters considered, as well as the final selected values, are shown in Supplementary Tables 2–8. Following the selection of the "best" hyperparameters within the inner loop, the model was then evaluated on the test set within the outer loop. This nested cross-validation strategy was implemented ten times for each ML model to determine the average model performance across randomly generated test sets. This cross-validation strategy resulted in some overlap for specific drugs between the inner and outer loops, as well as a similar overlap for specific polymers. However, grouping by drug–polymer combination ensured that there were no specific LAI systems (i.e., drug–polymer combinations) present in both the training/validation (i.e., inner loop) and test (i.e., outer loop) sets, thus allowing for cross-validation against drug–polymer based splits. In all cases, model performance was assessed using mean absolute error (MAE), the average absolute difference between predicted and experimental fractional drug release values.

The overall predictive performance of each ML model on the inner (i.e., training/validation) and outer (i.e., test) nested cross-validation loops are summarized in Table 1. The results obtained for each individual trial for all models are also shown in Supplementary Tables 9–12. Overall, the predictive performance of the outer loop largely reflected the respective accuracies found in the inner loop: indicating good model fitting. Specifically, the tree-based ML models were, on average, more accurate (MAE < 0.16) than the linear, instance-based, and deep learning models investigated. This is consistent with a recent study demonstrating that tree-based models remain state-of-the-art for medium-sized datasets (-10 K samples)[45]. The LGBM model had the highest overall prediction accuracy, with MAE values of 0.125 (±0.039) and 0.114 (±0.036) obtained for the inner and outer loops, respectively. Considering the MAE values across the various nested cross-validation trials, the performance of the LGBM model appeared to be closely matched by RF, NGB, and XGB models. To better illustrate the performance of these models, the absolute error values generated for all predictions across the ten nested cross-validation trials are summarized in Fig. 2. While the mean values for the LGBM, RF, NGB, and XGB models are close, the deviation of absolute error values varied between these forest-based models. Overall, the LGBM model afforded the most narrow distribution in absolute error values for the test data. There was found to be a statistically significant difference between the absolute error values generated by the LGBM model compared to the other models ($p < 0.05$). Thus, the LGBM model was selected for further development.

For comparison, we also trained and evaluated a series of ML models without any initial drug release measurements included as input features (i.e., without the features $T = 0.25$, $T = 0.5$, and $T = 0.1$). Models that include initial experimental measurements as input are referred to as few-shot models, and models without such inputs are referred to as zero-shot models. Few-shot models necessitate the measurement of the first few experimental points before the predictions can be performed; however, the result is often a more accurate model. In this study, it was found that the addition of initial drug release measurements (i.e., few-shot models) resulted in better performance than the corresponding models without these features (Supplementary Tables 9–13, and Supplementary Fig. 3).

### Model refinement
When trained using the 17 input features mentioned above, the LGBM model was identified as the most accurate at predicting fractional drug release. In general, input feature sets for ML models should be comprehensive and encode all relevant physical information to make accurate predictions. The use of too many, possibly irrelevant features

**Table 1 | Mean absolute error (MAE) values, as well as the standard error of the mean ($\sigma M$; shown in brackets), obtained using the different ML algorithms following prediction of fractional drug release during nested cross-validation (i.e., $n = 10$ trials)**

| | LGBM | RF | NGB | XGB | DT | NN | k-NN | SVR | lasso | PLS | MLR |
|---|---|---|---|---|---|---|---|---|---|---|---|
| The inner loop (validation) | 0.125 (0.039) | 0.130 (0.041) | 0.128 (0.040) | 0.133 (0.042) | 0.154 (0.049) | 0.166 (0.053) | 0.179 (0.057) | 0.206 (0.065) | 0.242 (0.076) | 0.246 (0.078) | 0.244 (0.081) |
| Outer loop (test) | 0.114 (0.036) | 0.119 (0.038) | 0.122 (0.039) | 0.126 (0.040) | 0.154 (0.049) | 0.184 (0.058) | 0.176 (0.056) | 0.248 (0.079) | 0.255 (0.081) | 0.252 (0.080) | 0.255 (0.081) |

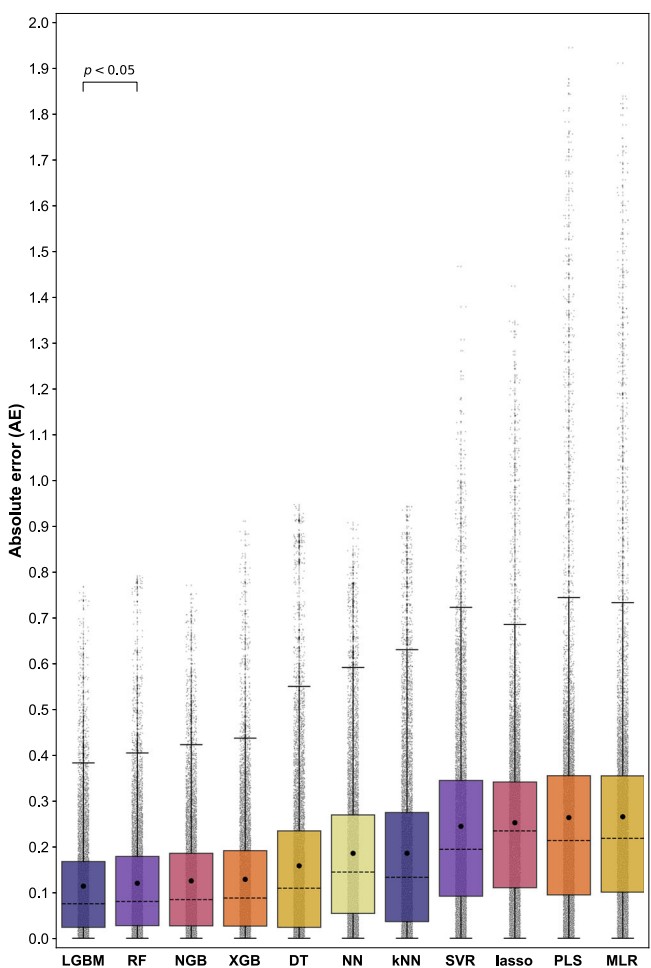

**Fig. 2 | Summary of the overall predictive performance of the various ML models represented as a box and whisker plot.** The data represent the absolute error (AE) obtained for fractional drug release predictions during nested cross-validation (i.e., $n = 10$ trials). Each column represents the AE value for 8013 data instances from the collective nested cross-validation test sets (light gray circles). The mean absolute error (MAE) and median AE values for each model are displayed within the boxes as closed black circles, and black dashed lines, respectively. The first and second quartiles are shown by the upper and lower edges of the respective boxes. The whiskers extend to show the rest of the distribution, excluding points determined to be "outliers" using the interquartile range method. Source data are provided as a Source Data file.

can increase the risk of introducing spurious correlations between individual features and observed release profiles which can eventually degrade the model's generalizability. Thus, the 17-feature LGBM model was assessed, refined, and optimized through agglomerative hierarchical clustering analysis to identify potentially redundant input features. The input features were arranged into a hierarchy of clusters using the farthest neighbor clustering algorithm (Fig. 3 and Supplementary Fig. 4). With this approach, strong correlations were observed between the input features *Drug_NHA*, *Drug_TPSA*, and *Drug_Mw* (>0.94), with *Drug_NHA* and *Drug_TPSA* displaying the strongest correlation (0.99). A similar cluster of strong correlations was observed between $T = 0.5$, $T = 0.25$, and $T = 1.0$ (>0.92), with $T = 0.5$ and $T = 1.0$ displaying the strongest correlation (0.97). Thus, there was some redundancy between the 17 features. The performance of the LGBM model was then assessed following the removal of select feature clusters based on their linkage distances (Table 2 and Supplementary Table 14). The optimal version of the LGBM model was determined to be the model that could achieve the lowest MAE while minimizing the number of features. This was identified to be a 15-feature LGBM model

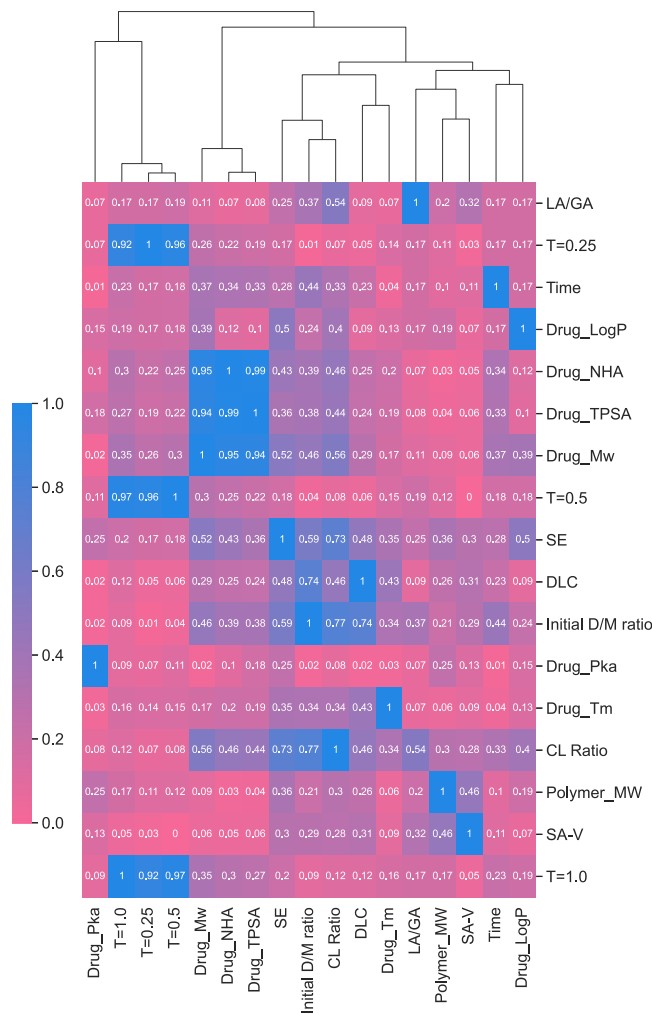

**Fig. 3 | Heatmap of the absolute Spearman's Rank correlation between the initial 17 input features.** Dark blue signifies an absolute Spearman's Rank correlation = 1, and pink represents an absolute Spearman's Rank correlation of 0. Attached to the heatmap is a dendrogram that displays the hierarchies of feature clusters that were determined via agglomerative hierarchical clustering analysis. Source data are provided as a Source Data file.

**Table 2 | Table summarizes the impact of feature cluster removal (i.e., based on their respective linkage distances) on the predictive performance of the LGBM model**

| No. of features | Mean absolute error (n = 10) | Standard deviation (n = 10) | Wards linkage distance |
|---|---|---|---|
| 17 | 0.116 | 0.018 | 0.00 |
| 15 | 0.116 | 0.017 | 0.06 |
| 13 | 0.142 | 0.017 | 0.12 |
| 12 | 0.143 | 0.017 | 0.24 |
| 11 | 0.143 | 0.018 | 0.29 |
| 10 | 0.143 | 0.019 | 0.35 |
| 9 | 0.139 | 0.022 | 0.53 |
| 8 | 0.150 | 0.021 | 0.76 |
| 5 | 0.296 | 0.023 | 0.82 |
| 4 | 0.296 | 0.024 | 0.88 |

The performance of the LGBM model with various numbers of input features was assessed by comparing the average and standard deviation of the AE values obtained from a series of trials (n = 10 trials) that randomly grouped 20% of the drug–polymer combinations as a holdout test set.

that did not include *Drug_NHA* or *T* = 0.5. Interestingly, despite the strong correlation between *Drug_TPSA*, and *Drug_Mw* as well as between *T* = 0.5 and *T* = 1.0 it was found that the removal of any additional feature clusters resulted in a reduction in model accuracy (Table 2). In the case of *Drug_NHA* and *Drug_TPSA*, both values were generated by RDKit (based on SMILES codes), and these predictions may be incorrect and, therefore, irrelevant to the final model. For the *T* = 0.5 and *T* = 1.0 features, there may not be a sufficient number of fractional drug release values with meaningful differences at these time points. This final 15-feature LGBM model was then analyzed and deployed in a prospective study. To confirm that suitable hyperparameters were selected for the 15-feature LGBM, the nested cross-validation strategy was conducted again with this reduced number of input features. The performance was consistent with that of the 17-feature model, while the resulting hyperparameter configuration differed slightly.

## Model interpretation

To illustrate the predicted fractional drug release profiles generated by the trained LGBM model, test set drug–polymer combinations were extracted from the nested cross-validation and plotted against their respective experimental drug release profiles (Fig. 4 and Supplementary Fig. 5). Figure 4a shows the predicted and experimental drug release profiles for select LAIs from the dataset. Overall, there was good agreement between the predicted and experimental drug release profiles. The importance of the various input features in generating such fractional drug release predictions was then determined via Shapley additive explanation (SHAP) analysis of the final 15-feature LGBM model. Figure 4b shows the global contribution of each feature in terms of SHAP values. In Fig. 4b, the input features are arranged from top to bottom in order of decreasing the impact on model output. *Time* is listed in the top row, indicating that it had the greatest impact on the fractional drug release predictions. In addition, each row of the swarm plot in Fig. 4b also depicts an individual dot plot, with each dot representing a separate instance in the dataset. Blue dots indicate lower values for that feature in the dataset, while pink dots represent higher values. For example, blue dots in the *Time* row represent low values for drug release timepoints, while pink dots in the *Polymer_MW* row represent instances for LAIs composed of polymers with higher molecular weights in the dataset. The distance between an individual dot and the gray vertical central line reflects the extent to which that fractional drug release value deviates from the mean value of the initial dataset. For example, lower *Time* values (i.e., blue dots in the *Time* row, Fig. 4b) resulted in a strong negative contribution to fractional drug release, while higher *Time* values (i.e., pink dots in the *Time* row, Fig. 4b) resulted in a strong positive contribution. After *Time*, the next most influential features were *T* = 1.0, *Drug_Mw*, and *Polymer_MW*. *T* = 1.0 was observed to have a similar relative contribution to fractional drug release as *Time*. Specifically, lower values of *T* = 1.0 resulted in a strong negative contribution to fractional drug release, while higher values resulted in a strong positive contribution. Interestingly, two of the most important features are descriptors of the physicochemical properties of the drug and polymer (i.e., *Drug_Mw*, and *Polymer_MW*). This indicates that, of the components of the LAI system, the LGBM model has recognized that the molecular weight of the drug and the molecular weight of the polymer has the most significant influence on fractional drug release. The remaining molecular and physicochemical descriptors appear to have minimal contribution to fractional drug release prediction for this model. However, it should be noted that Fig. 4b only shows the effect of each individual feature and does not account for potential synergy between input features. Moreover, we know from the agglomerative hierarchical clustering analysis (Fig. 3) that a reduction in any of the remaining features has a negative impact on the performance of the model.

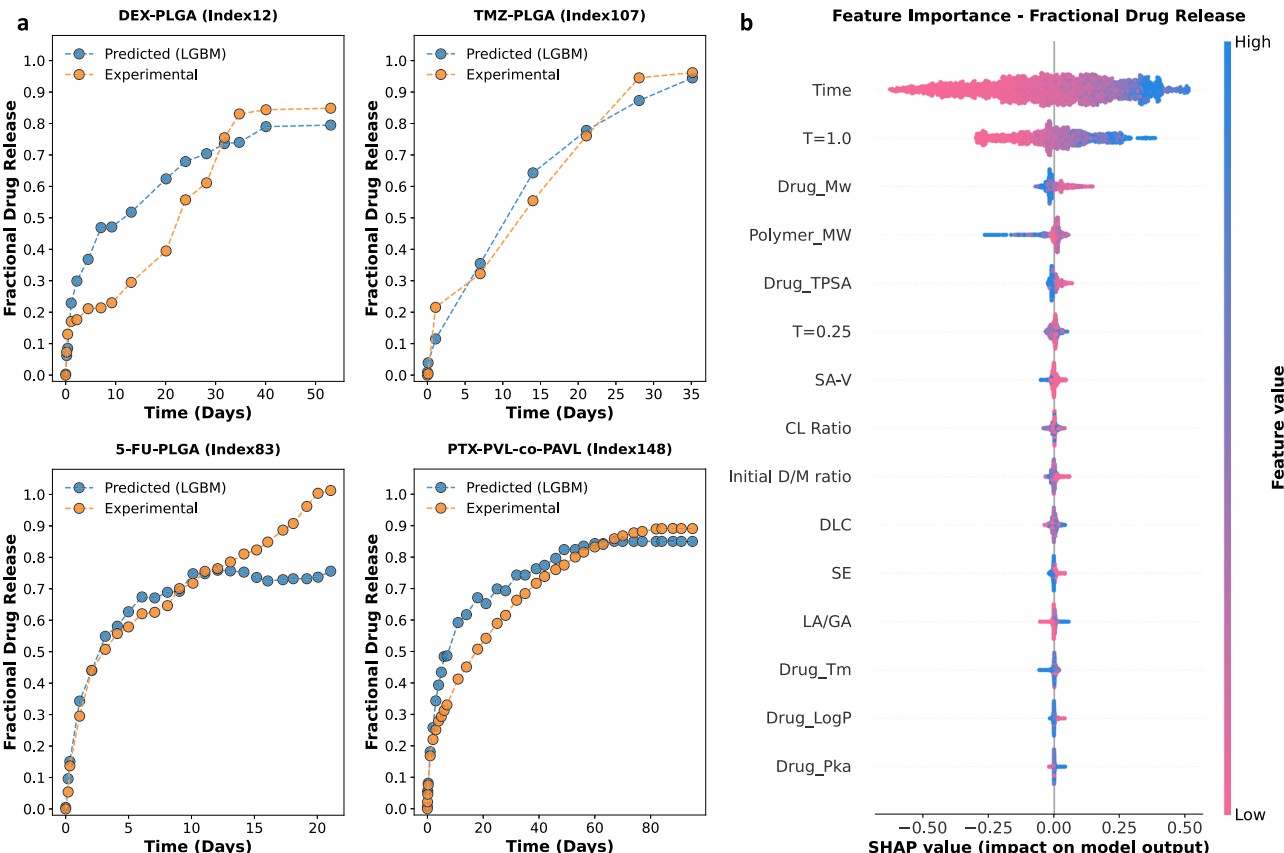

**Fig. 4 | Deployment of the trained 15-feature lightGBM (LGBM) model. a** Select examples of experimental fractional drug release profiles (orange circles) in comparison to predicted fractional drug release profiles (blue circles) generated by the LGBM model. These include dexamethasone-loaded PLGA MPs (DEX-PLGA); temozolomide-loaded PLGA MPs (TMZ-PLGA); fluorouracil-loaded PLGA MPs (5-FU-PLGA); and paclitaxel-loaded PVL-co-PAVL cross-linked cylinders. **b** Shapley additive explanations (SHAP) analysis for the 15-feature LGBM model. The impact of each feature on fractional drug release is illustrated through a swarm plot of their corresponding SHAP values. The color of the dot represents the relative value of the feature in the dataset (high-to-low depicted as pink-to-blue). The horizontal location of the dots shows whether the effect of that feature value contributed positively or negatively in that prediction instance (x-axis). Source data are provided as a Source Data file.

The decision path function from the SHAP library was utilized to better visualize the collective effect of the 15 input features on model outputs. This type of analysis highlights the combined effect of the input features and allows for individual data instances to be followed and interpreted. Figure 5 illustrates how the LGBM model determines fractional drug release predictions, using a single LAI formulation from the dataset. 5-FU-PLGA (index 84 in the attached dataset) is a spherical MP (~30 μm) formulation of fluorouracil (L/G = 50:50, Mw = 104 kDa) with a DLC of 18% (wt%). Figure 5a shows a visual comparison of the experimental and predicted fractional drug release values for 5-FU-PLGA (index 84 in the attached dataset) and highlights three arbitrary prediction instances (i) *Time* = 0 days, predicted fractional drug release (i.e., $f(x)$) = 0.01; (ii) *Time* ≃ 7 days, $f(x)$ = 0.61; and (iii) *Time* ≃ 20 days, $f(x)$ = 0.82. Figure 5b shows the decision path for each of the predicted fractional drug release values. Each prediction begins at the base value for the model and migrates through the LGBM model based on the contributions of each input feature. Figure 5c shows a breakdown of the exact contribution of each input feature on the prediction of fractional drug release for the three highlighted data instances, as SHAP force plots. In Fig. 5c, the relative values of each input feature are shown by a pink (positive) or blue (negative) band on the force plot, with the width of this band representing the numerical contribution to the final model output. The contributions of each input feature are summed with the base value of the model to derive the model output.

## Prospective study

Utilizing SHAP analysis allowed us to establish the importance ranking of the 15-feature LGBM model (Fig. 4) and to better understand their combined contributions in generating fractional drug release predictions (Fig. 5). This analysis also revealed that the input features which had the greatest influence on the fractional drug release predictions were *Time* and *T* = 1.0. While these are certainly features that reflect the performance of an LAI, they are not known by formulation scientists a priori. This information only becomes available once the LAIs have been prepared and characterized. However, the importance of *Time* and *T* = 1.0 to the model is intuitive, given that *Time* is the only feature of the model that varies for a given LAI system. *T* = 1.0 provides the model with fractional drug release values after one day and could be considered a proxy for the initial drug release rate. For instance, low values of *T* = 1.0 are likely to be indicative of sustained release profiles compared to LAIs with high fractional drug release at *T* = 1.0. Thus, using *T* = 1.0 as a proxy for initial drug release rates might inform the design attributes of new LAIs. We were interested in determining the impact of input features that are more easily manipulatable by formulation scientists (e.g., drug properties, polymer types, LAI morphology, etc.). To investigate this, we used a simple unsupervised clustering approach (i.e., T-distributed stochastic neighbor embedding; t-SNE) to group the input features of the LGBM model by their respective SHAP values, Supplementary Fig. 6.

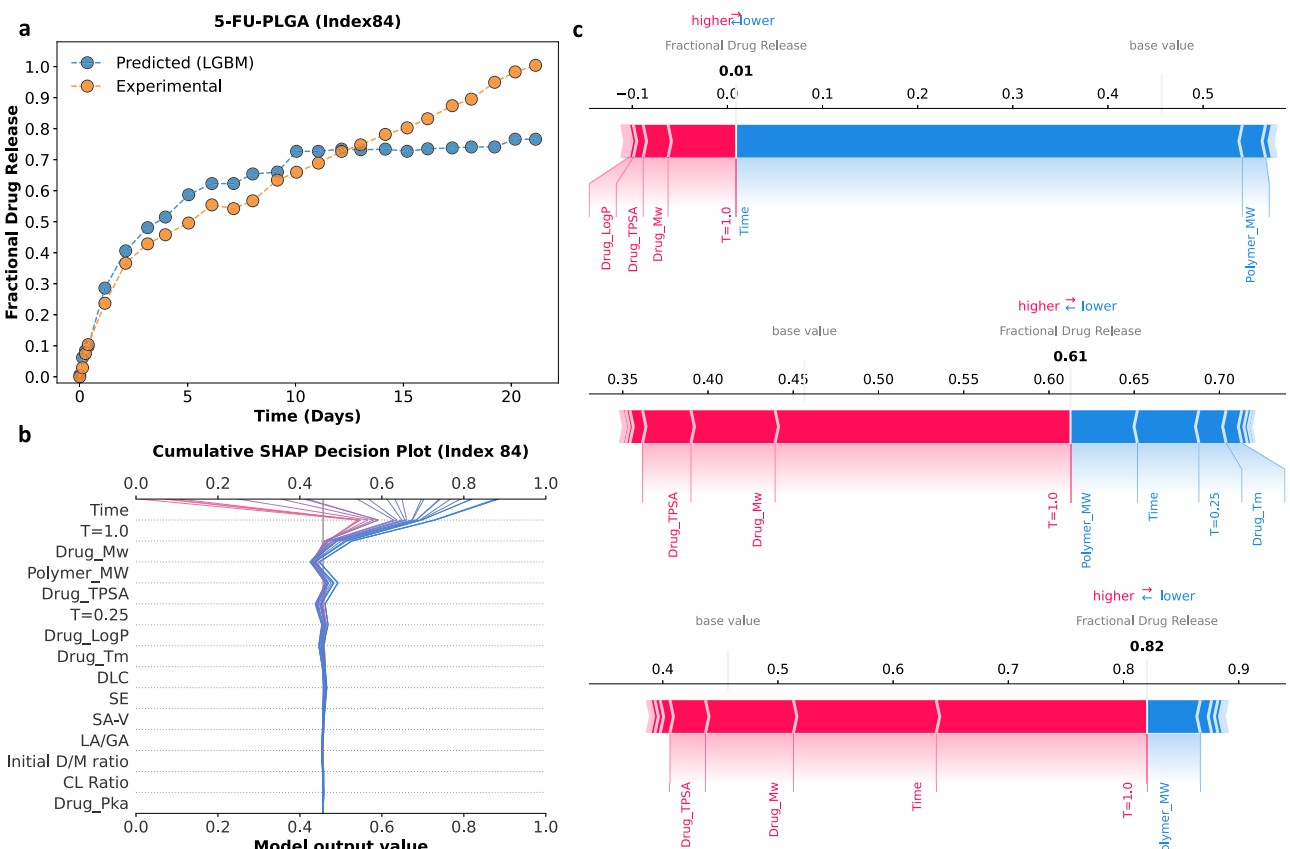

**Fig. 5 | A visual representation of how the 15-feature LGBM model generates fractional drug release predictions using the example of 5-FU-PLGA (index 84 in the attached dataset). a** Experimental fractional drug release profile (orange circles) for 5-FU-PLGA plotted against the predicted fractional drug release profile (blue circles) generated by the LGBM model. **b** Decision path taken for each fractional drug release prediction for the 5-FU-PLGA system. This plot illustrates how the LGBM model combines the relative contribution of each input feature to return the predicted fractional drug release value. **c** Shapley additive explanations (SHAP) force plots for the three selected data instances (i.e., fractional drug release prediction 0.01, 0.61, and 0.82) showing a decomposition of predicted fractional drug release values into the relative SHAP contribution values for each input feature. The relative SHAP values for each input feature are shown by pink (positive) or blue (negative) bands on the force plot, with the width of the band representing the numerical contribution to the final model output. Source data are provided as a Source Data file.

With the assumption that lower drug release values at $T=1.0$ are indicative of an LAI that provides sustained drug release (and vice versa), it is possible to identify other features of the LAI systems that are correlated with low values of $T=1.0$. This was done by highlighting relative feature values in the low-dimensional cluster map of the dataset and comparing across features. Select examples of $T=1.0$, *CL_Ratio*, *Polymer_MW*, and *Drug_Mw* are shown in Fig. 6. This information can be used to guide the design of new LAI systems. As a starting point, we focused on the design of PLGA-based MPs. PLGA was selected for the preparation of the formulations due to its commercial availability (including a wide range of molecular weights), relatively straightforward methodology for MP production (compared to cross-linked systems), and widespread use in commercially available LAI products[8]. Thus, our design criteria were immediately limited to systems with *CL_ratio* values equal to zero to exclude any of the cross-linked particles or cylinders in the collected dataset. This subset of the dataset can be seen by the pink circles (i.e., low values) in the CL Ratio subplot of Fig. 6. Following this subset of the data to the other subplots, it was noted that certain feature values were loosely associated with drug release at $T=1.0$, and therefore indicative of whether an LAI would release drug faster or slower. For instance, medium-high values for *Polymer_MW*, medium-high values for *Drug_Mw*, and low-to-medium values of DLC were generally associated with low values for $T=1.0$ for the PLGA LAIs. While lower values for *Polymer_MW*, low values for *Drug_Mw*, and low values for DLC were generally associated with high

values for $T=1.0$, for the PLGA LAIs. These trends are further highlighted in Supplementary Fig. 7, with the inclusion of an expanded series of input features. Thus, based on this analysis, we devised a table of key design criteria for PLGA MPs (Table 3). It should be noted that the relative values for these features are constrained to the values described in the initial dataset that was used to train and evaluate the model.

Based on the results of the SHAP analysis, we proceeded to prepare and characterize two separate LAIs which fulfilled the design criteria identified for fast-release and slow-release PLGA formulations (Table 3). Specifically, for the "fast" release LAI, which required a relatively low *Polymer_MW*, a 10 kDa PLGA was selected. While for the "slow" release LAI, which required a medium-to-high Polymer_MW, a 50 kDa PLGA was selected. The drugs to be encapsulated in the respective polymers were selected based on two criteria (i) physicochemical properties in the range determined by SHAP analysis and (ii) exclusion from the dataset used to train the model. Specifically, salicylic acid (SA) was selected as the drug for the "fast" release PLGA formulation due to its relatively low values for *Drug_Mw* (138.12 g/mol), *Drug_TPSA* (57.53 Å), *Drug_pKa* (1.79), *Drug_logP* (1.09), and *Drug_Tm* (159.6 °C). Olaparib (OLA) was selected for the "slow" release PLGA-based formulation due to its relatively high values for *Drug_Mw* (434.47 g/mol), *Drug_TPSA* (86.37 Å), *Drug_pKa* (9.96), *Drug_logP* (2.35), and *Drug_Tm* (208.5 °C).

The LAIs were then prepared using a well-established emulsion-based method at drug-to-material ratios such that the values for DLC

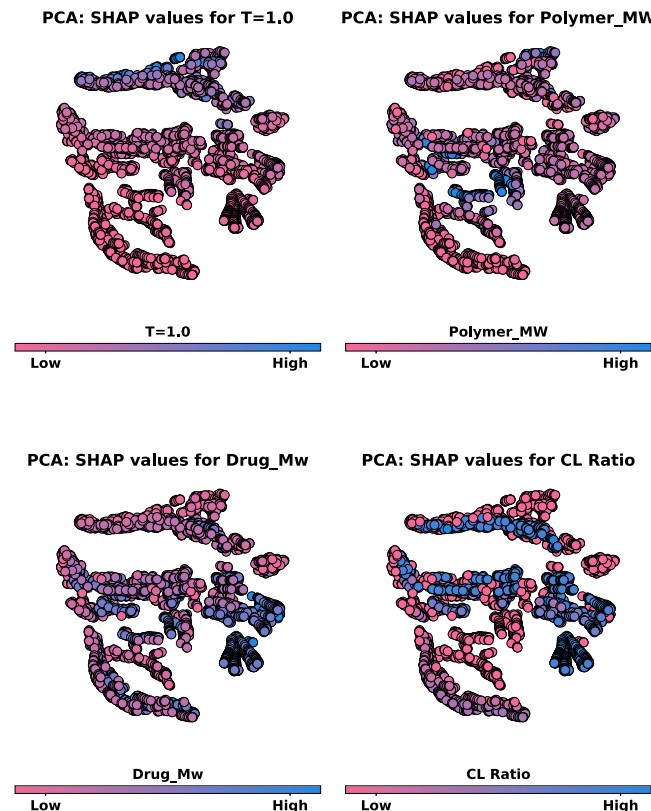

**Fig. 6 | Dimensionality reduction combined with Shapley additive explanations (SHAP) analysis.** Two-dimensional visualization of the SHAP values calculated for the input features of the LGBM model. The SHAP values for the 15 input features were condensed into two principal components using principal component analysis (PCA) and then grouped together using a simple unsupervised clustering algorithm (T-distributed Stochastic Neighbor Embedding). This low-dimensional/clustered plot was then utilized to visualize and compare the location of data instances corresponding to different input features, including $T = 1.0$; $CL\_Ratio$; $Polymer\_MW$; and $Drug\_Mw$. In each case, the attached colorbar depicts the relative value of that feature in the dataset ranked from high (blue) to low (pink). Source data are provided as a Source Data file.

**Table 3 | Table highlighting key design criteria for "fast" and "slow" release PLGA MPs based on SHAP analysis of the 15-feature LGBM model and the observed trends in the PCA plots**

| Feature/property | Value for "fast" release | Value for "slow" release |
|---|---|---|
| Drug MW | Low | Medium/high |
| Polymer MW | Low | Medium/high |
| Drug TPSA | Low | Low/medium |
| SA-V | Low | Low/medium |
| Initial D/M ratio | Low | Low |
| DLC | Low | Low/medium |

(and *Initial D/M ratio*) were within the LGBM model's design criteria for the "fast" and "slow" release formulations. Specifically, the DLC was <0.1 for the "fast" release formulation and >0.2 for the "slow" release formulation. For both the "fast" and "slow" systems, the values obtained for *SA-V* are indicative of spherical particle morphologies. However, the "slow" release LAIs had some medium dataset values, thus, the OLA-PLGA system was prepared with a larger particle size than the SA-PLGA. The physicochemical properties of the OLA-PLGA and SA-PLGA LAIs are summarized in Supplementary Table 15. The average particle size of the MPs was determined using laser diffraction

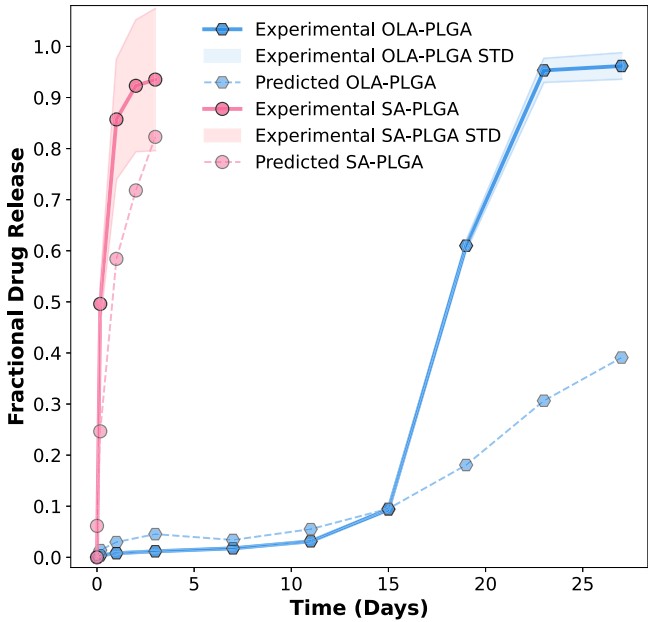

**Fig. 7 | Comparison of the experimental and predicted fractional drug release profiles for the salicylic acid-PLGA MP (SA-PLGA) and olaparib-PLGA MP (OLA-PLGA) formulations.** The design of both the SA-PLGA and OLA-PLGA was based on SHAP analysis of the trained LGBM model. Three independent batches of both formulations were prepared, and their experimental drug release was characterized in 0.5 wt% sodium dodecyl sulfate (SDS) to ensure sink conditions throughout the experiments. The fractional experimental drug release profiles for SA-PLGA and OLA-PLGA are plotted together as pink circles with a solid line (SA-PLGA) and blue hexagons with a solid line (OLA-PLGA), respectively. In both cases, the standard deviation (STD) observed for these experimental drug release measurements is displayed as a colored halo ($n = 3$). The fractional drug release profiles predicted by the LGBM model are also shown as pale pink circles with a broken line (SA-PLGA) and pale blue hexagons with a broken line (OLA-PLGA), respectively. Source data are provided as a Source Data file.

particle size analysis, and the morphology was evaluated using scanning electron microscopy (Supplementary Fig. 8). Once prepared and characterized, the in vitro release profiles of drugs from the LAIs were evaluated in 0.5 wt% SDS and the resulting experimental release profiles were compared with the LGBM model predictions. The results of this prospective study are shown in Fig. 7 (Supplementary Fig. 9). Overall, there was good agreement between the predicted and experimental release profiles for both the "fast" and "slow" release PLGA systems. However, it was noted that the experimental release of OLA-PLGA increased significantly after day 15 compared to the predicted release profile. This increase in experimental release rate is most likely the result of PLGA hydrolysis in the release media, which is known to result in a change in release rate[46]. This highlights the need for more data on slow-release PLGA-based formulations and/or integration of information related to the time-dependent hydrolysis of PLGA into the model.

Despite this potential limitation of the current model, the results for the OLA-PLGA system are promising. OLA is a poly ADP ribose polymerase (PARP) inhibitor that is commercially approved (Lynparza®) for the treatment of advanced ovarian cancer, metastatic prostate cancer, metastatic pancreatic cancer, and early (or metastatic) breast cancer[47]. The recommended dose of Lynparza is 300 mg (two 150 mg tablets) taken orally twice a day, and treatment continues until disease progression or unacceptable toxicity is observed[48]. Thus, OLA is an example of a therapeutic agent that could potentially benefit from the formulation as an LAI. While it remains to be seen if additional optimization is required to arrive at a version of OLA-PLGA that is efficacious in vivo, the deployment of the LGBM

model here has directed the design of an LAI (OLA-PLGA) with significant potential.

In summary, cutting-edge ML technologies are now freely available to pharmaceutical and materials scientists. The results obtained in this study demonstrate the potential for ML to expedite the development of innovative drug-delivery technologies. Among the strengths of modern ML techniques are their ability to provide insights into how models reach their predictions. Herein we demonstrate that ML models can not only be used to predict in vitro drug release from LAIs with a high degree of accuracy but also that interpretation of such models can be used to guide the design of new formulation candidates. In the current study, we found that for this dataset, the tree-based LGBM model provided the most accurate prediction of fractional drug release. Given the small size of the dataset (<4000 observations) and that most of the data points contain variables that are properties of the drug or polymer, it is perhaps not surprising that the neural network models investigated did not perform well. As the use of ML in drug formulation development increases, we anticipate that larger datasets will become available, leading to an increase in the utility of neural networks. In the meantime, the implementation of tree-based models such as LGBM has the potential to reduce the time and cost associated with the development of LAI formulations. For instance, the data-driven approach used here directs the design of a promising LAI for the drug olaparib (OLA-PLGA). However, it remains to be seen if additional optimization is required to arrive at a version of OLA-PLGA that is efficacious in vivo. Overall, these results demonstrate a promising application of ML in drug formulation development. It is our hope that this proof-of-concept study, and its associated dataset, will aid in fostering the development of more advanced, tailored, and accurate ML approaches for the design of new drug formulations.

# Methods

## Machine learning

**Data collection.** The dataset was constructed from previously published studies by our group and other research groups[10,17–44]. The studies performed by our group include spherical and cylinder-shaped polymeric LAIs[10,18,38]. Data from external sources were identified using the Web of Science search engine and the keyword combination "polymeric microparticle" and "drug delivery"[17,19–37,39–44]. Information related to the preparation, final composition, and release kinetics of drugs from LAIs was collected. The latter was primarily extracted from figures of in vitro drug release profiles using the "GetData Graph Digitizer" application. The final dataset contained 181 drug release profiles for 43 unique drug-polymer combinations. In total, this comprised 3783 individual fractional release measurements. The initially collected dataset was composed of a table of drug and polymer names, as well as physicochemical properties of the formulation, and fractional drug release values at various timepoints. In order to use this data to construct and train ML models, it is necessary to describe various elements using machine-readable descriptors which were generated using RDkit. The polymers and LAI formulations were described exclusively using information reported in the relevant published articles; these included; *polymer_MW*, *LA/GA* (for non-PLGA systems, this was set as zero), *molecular CL_Ratio* (for non-cross-linked systems, this was set as zero), initial *D/M* ratio, *DLC*, SA-V ratio for the LAI system, fractional drug release at 6 h ($T = 0.25$), fractional drug release at 12 h ($T = 0.5$), fractional drug release at 24 h ($T = 1.0$), and the percent of surfactant present in the release media (*SE*; where no surfactant was present in the release media, this was set as zero). With the exception of *SA-V*, $T = 0.25$, $T = 0.5$, and $T = 1.0$, the 17 input features were either extracted from original publications or calculated using the RDkit package. *SA-V* was constructed and implemented for this study as it confers information that is related to the size and shape of the LAI system. This enables the inclusion of both spherical and cylindrical-shaped LAIs in one model. For studies wherein values

for fractional drug release were not available at early timepoints (i.e., $T = 0.25$, $T = 0.5$, and $T = 1.0$) they were imputed using best-fit polynomial curves that range from $T = 0$ to $T = 2$ days.

**Data splitting strategy for ML model training.** ML models were trained and evaluated using a nested cross-validation approach. The nested cross-validation approach consisted of an inner (training and validation) and outer (test) loop. To implement the nested cross-validation approach, the collected dataset was grouped by drug-polymer combinations (43 unique drug–polymer combinations in total), to allow cross-validation against drug–polymer based splits. For each ML model, ten trials of the nested cross-validation method were conducted to assess the general performance of each algorithm. For each implementation of the nested cross-validation method, 20% of the drug-polymer groups in the dataset were randomly held back for evaluation using the group-shuffle-split method from the Scikit-learn library in Python[49]. The remaining 80% of the drug-polymer groups were then used for hyperparameter selection of models in the inner loop. This was done using the random grid search method from the Scikit-learn library in Python[49], with 100 random hyperparameter configurations being assessed in each case. Within the inner loop, data were again grouped and split by drug–polymer combination using the Group k-fold cross-validation strategy from the Scikit-learn library in Python[49] (where k = 10). Thus, 10% of the drug-polymer groups within the inner loop were used for hyperparameter evaluation, while the other 90% were employed to train the various model architectures selected using the random grid search method. The outer loop was used to test the performance of the model structure (i.e., fixed hyperparameters) selected by the inner loop using a holdout dataset (i.e., 20% of the drug-polymer combinations in the initial dataset).

**ML model development.** In total eleven ML algorithms were trained and investigated for this task. These included MLR, lasso, PLS, DT, RF, LGBM, XGB, NGB, SVR, k-NN, and NN models. All of these models were built and evaluated in Python. NN models were built using the Keras package with the backend of TensorFlow[50], LGBM models were built using the lightGBM package[51], XGB models were built using the XGBoost package[52], NGB models were implemented using the NGBoost package[53], and all other models were built using the Scikit learn library[49]. In all cases, prior to training any of the ML models, a data preprocessing step was conducted to standardize the data prior to input into the ML models. This was done using the standard scalar package available in the Scikit learn library[49]. ML model hyperparameters were tuned using the randomized grid search package in Scikit learn[49], and the negative mean absolute error metric was employed.

**ML model evaluation.** To assess the predictive performance of all trained ML models, the predictions for each outer loop ($n = 10$) split from the nested cross-validation were compiled into a data frame for analysis. This amounted to ~8000 data samples and enabled a more quantitative evaluation of general prediction accuracy for each model. This was done by determining the absolute error (AE) for each prediction made for each sample in each of the ten outer loops. The AE and MAE were determined using the equations shown below

$$\text{Absolute error (AE)} = |\hat{y}_i - y_i| \tag{1}$$

$$\text{Mean absolute error (MAE)} = \frac{\sum_{i=1}^{n} |\hat{y}_i - y_i|}{n} \tag{2}$$

Where $\hat{y}_i$ is the predicted fraction drug release value; $y_i$ is the experimental fractional drug release value obtained from either the training or external validation datasets; $n$ is the total number of data points.

**Feature engineering.** Agglomerative hierarchical clustering analysis was performed using the hierarchical clustering package from SciPy in Python[54] to arrange the initial input features into a hierarchy of clusters using the farthest neighbor clustering algorithm. The performance of the optimal ML algorithm (LGBM) was then assessed following the removal of select clusters based on their linkage distance. Here, the hyperparameter structure for the LGBM model identified during preliminary screening (i.e., nested cross-validation) was utilized, and only the number of input features was varied.

**Model interpretation.** SHAP analysis was conducted on the trained 15-feature LGBM model. The effect of the various input features on the fractional drug release prediction for the initial dataset was assessed using the TreeSHAP package and the force plot visualizations from the SHAP library in Python[55,56].

**Principal component analysis.** Dimensionality reduction was necessary to better visualize the impact of the model input features on initial drug release values. We used t-SNE to cluster the input features of the LGBM model by their respective SHAP values. To employ the t-SNE algorithm, we first utilized PCA to reduce the dimensionality of the 15-feature LGBM dataset into two principal components.

## Prospective study

**Materials.** Resomer® RG 504 H (PLGA, LA:GA = 50:50, Mw = 38–54 k), Resomer® RG 502 H (PLGA, LA:GA = 50:50, Mw = 7–17 k,), salicylic acid (SA, ≥99.0%), poly(vinyl alcohol) (PVA, Mw = 13–23 k, 87–89% hydrolyzed), sodium dodecyl sulfate (SDS, ≥98.5%), and dimethyl sulfoxide (DMSO, ≥99.9%) were purchased from Sigma Aldrich (ON, CA). Acetonitrile (ACN, HPLC grade) and formic acid (FA, LC/MS grade) were purchased from Fisher Chemical (ON, CA). Dichloromethane (DCM, HPLC grade) and methanol (HPLC grade) were purchased from Caledon Laboratories Ltd. (ON, CA). Olaparib (OLA, ≥99%) was purchased from MedKoo Bioscience (NC, US). Phosphate buffered saline (PBS, Gibco, pH = 7.4) was purchased from Thermofisher (Massachusetts, USA). Tween 80 (reagent grade) was purchased from Bio-Shop (ON, CA).

**Formulation preparation.** PLGA MPs were prepared based on a modified version of a previously reported oil-in-water (o/w) emulsion method[25]. For OLA MPs, 300 mg of RG 504 H and 200 mg of OLA were dissolved in 6 mL of a solvent mixture (DCM:DMSO = 50:50, v/v) in a 20 mL scintillation vial to produce the organic phase. The organic phase was then emulsified in 200 mL of MilliQ water (0.01 wt% PVA, previously cooled to 4 °C) using a homogenizer (L5M-A Lab Mixer, Silverson, MA, US) at 1500 rpm for 30 min. After homogenization, the emulsion was added to 500 mL of MilliQ water (without PVA, previously cooled to 4 °C) and stirred at 500 rpm for 60 min. The MPs were then filtered using a 100 μm cell strainer (Fisherbrand Sterile Cell Strainers, ON, CA), followed by a 70 μm cell strainer (Fisherbrand Sterile Cell Strainers, ON, CA) to retain the MPs within the size cut-off (i.e., 70–100 μm). MPs were then collected and frozen at −80 °C for 30 min prior to lyophilization overnight. For SA MPs, 300 mg of RG 502 H and 150 mg of SA were dissolved in 4 mL of a solvent mixture (DCM:DMSO = 75:25, v/v) in a 20 mL scintillation vial to produce the organic phase. SA MPs were then formulated as described for the OLA MP formulation, except for the filtration step, where 70 μm and 40 μm cell strainers (Fisherbrand Sterile Cell Strainers, ON, CA) were utilized to retain the MPs with diameters ranging from 40 to 70 μm.

**Size measurement.** Measurements of particle size were conducted via laser diffraction, using a Mastersizer 300 (Malvern, ON, CA) equipped with a Hydro SV accessory. For both OLA and SA MPs, around 5–10 mg of lyophilized MPs were suspended in 100 μL of MilliQ water (with

0.2 wt% Tween 80). The 100 μL MP suspensions were then added to the system for evaluation.

**Morphology.** The morphologies of lyophilized OLA and SA MPs were evaluated using scanning electron microscopy (SEM, Quanta FEG 250 ESEM) at the Centre for Nanostructure Imaging at the University of Toronto.

**Drug loading analysis.** To evaluate drug loading, ~10 mg of lyophilized MPs were weighed, and the drug was extracted using 4 mL of a DCM:DMSO solvent mixture (50:50 for OLA and 75:25 for SA, v/v) in a 20 mL scintillation vial. This solution was then vortexed until the MPs were fully dissolved in the solvent. Following this, the resulting solution was filtered (0.45 μm, Millex-HV PVDF Syringe Filters, Sigma Aldrich, CA) and appropriately diluted in methanol prior to drug quantification via high-performance liquid chromatography (HPLC). HPLC analysis of OLA and SA was performed using an Agilent Technologies 1260 Infinity II (Agilent Technologies, Santa Clara, CA, USA) with a photodiode array (PDA) detector and an XDB-C18 column (4.6 × 150 mm, ZORBAX Eclipse, Agilent). The mobile phase was composed of MilliQ water:ACN (70:30 wt%, with 0.1 wt% FA), and the flow rate was 1.2 mL/min. OLA and SA were analyzed at wavelengths of 254 nm and 303 nm, respectively. The experimental drug loading capacity (i.e., DLC) was calculated as follows:

$$\text{Drug loading capacity (\%)} = \frac{\text{Mass of drug extracted}}{\text{Mass of drug loaded MPs}} \times 100\% \quad (3)$$

**Drug solubility in the release media.** To ensure the in vitro drug release studies were conducted under sink conditions, the solubility of the drug in the release media (PBS with 0.5 wt% SDS) was measured. Specifically, an excess amount of drug (~10 mg of OLA and ~30 mg of SA) was added to 5 mL of release media (i.e., PBS with 0.5% Tween) in scintillation vials. The scintillation vials were incubated at 37 °C with gentle stirring (50–100 rpm) overnight, followed by filtration using a filter membrane (0.45 μm, Millex-HV PVDF Syringe Filters, Sigma Aldrich, CA). The drug concentrations were then quantified using HPLC as described in the previous section.

**Drug release assay.** To characterize the in vitro drug release profiles, 8–10 mg of OLA MPs (corresponding to ~2.5 mg of drug) and 20–40 mg of SA MPs (corresponding to ~0.5 mg of drug) were weighed for each release assay. The MPs were then suspended in centrifuge tubes (15 mL, Corning, Sigma Aldrich, CA) containing 15 mL of release media. The tubes were incubated at 37 °C with gentle mixing using a tube shaker (Labquake, Fisher Scientific). At predetermined timepoints, the release samples were centrifuged at 45 × g for 3 min to pellet MPs at the bottom of the tube. Aliquots (1 mL) of the supernatant were analyzed, the remaining media was discarded, and fresh release media (15 mL) was added to each tube to ensure sink conditions. Drug release was quantified using HPLC as previously described. Fractional drug release was calculated as follows:

$$\text{Fractional drug release} = \frac{\text{Mass of drug released}}{\text{Mass of drug loaded}} \quad (4)$$

## Reporting summary

Further information on research design is available in the Nature Portfolio Reporting Summary linked to this article.

## Data availability

The dataset and results that support the findings of this study are available on Zenodo and ChemRxiv. Source data are provided in this paper.

## Code availability

The code that supports the findings of this study is available at the Aspuru-Guzik Group's GitHub page and on Zenodo[57].

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

## Acknowledgements

NSERC Discovery grant (RGPIN-2022-04910) to C.A. F.H., M.A., and A.A.G. acknowledge support from the Defense Advanced Research Projects Agency under the Accelerated Molecular Discovery Program under Cooperative Agreement No. HR00111920027 dated August 1, 2019. R.J.H. gratefully acknowledges NSERC for a postgraduate scholarship (PGSD3-534584-2019), as well as support from the Vector Institute. A.A.G. would like to thank Dr. Anders Frøseth for his support. M.A. is supported by a postdoctoral fellowship from the Vector Institute. Figure 1 was created with BioRender.com.

## Author contributions

P.B. co-led the conceptualization of this work, led the computational studies, led the data analysis, and wrote the first draft of the paper. Z.B. led dataset construction, assisted with the computational studies, assisted with data analysis, performed the experimental studies, and assisted with the review/editing of the paper. R.J.H. assisted with the computational studies, assisted with data analysis, and assisted with the review/editing of the paper. M.A. assisted with the computational studies, assisted with data analysis, and assisted with the review/editing of the paper. F.H. assisted with the computational studies, assisted with data analysis, and assisted with the review/editing of the paper. A.A.G. co-led conceptualization of the work, assisted with supervising the work, acquired research funding and assisted with review/editing the paper. C.A. co-led conceptualization of the work, led the supervision of the work, acquired research funding, and led the review/editing of the paper.

## Competing interests

The authors declare no competing interests.
