## [Peer Review File · Nature Communications]

Machine Learning Models to Accelerate the Design of Polymeric Long-Acting InjectablesREVIEWER COMMENTS

Reviewer #1 (Remarks to the Author):

Machine Learning Models to Accelerate the Design of Polymeric Long-Acting Injectables

Summary: The authors present their work on training machine learning (ML) models to predict the efficacy of long-acting injectables. The ML models are trained

Methodology:

Page 5, line 119. "from previously published studies". Please add the references to the studies the first time they are mentioned. Was this data generated by the same research group? I would also appreciate a summary of what they contain.

What happens if the models are trained on the current external validation dataset and evaluated on the current training data?

The majority of the variables used are properties of the molecule that do not change (I believe only release and time does). Since the predictions are made over time, I feel the authors should explain which variables are constant and which ones aren't.

Quality of the figures in the supplement is poor. Please update the supplement with high quality figures so that they are readable.

Was any of the drugs in the external dataset present in the training set? This is not mentioned and should be clarified.

Can the authors provide a tsv file instead of the excel files (same matrix employed to train the ML models)? Ideally, it should be released in zenodo to be used as a benchmark for future studies. The GitHub repo should be restructured so that there are directories (Python modules) where all the models are located.

Page 7 line 62. "As well as" shouldn't be preceded by a comma.

The validation proposed by the authors is strong but it could be improved if the authors conduct a nested-CV approach and report the values of the test dataset as well. Then the best hyperparameters would be tested in the external dataset (Figure 1). It would be important to show that the most optimal hyperparameters match what has been reported now with a single CV.

Figure 1 bar for MLR goes out of the plot.

Given the small amount of data points, it is not surprising that NN do not perform well here. Also, given that most of the data points contain variables that are properties of the chemical, tree-based models (using thresholds) are likely to be the best performing models. This should be discussed in the conclusion. Also, which other variables could be added and why they couldn't be added in the current study.

Do drug_TPSA and Durg_NHA values come from Rdkit predictions or were collected from studies? Maybe the predictions are wrong and that's why they are irrelevant.

Conclusion

Why is it hard to collect more data points than the current ones? In other words, why are there only a few drugs that have been used for training the ML models and not over 100 for instance?

Reviewer #3 (Remarks to the Author):

The authors present different machine learning approaches for the design of long-acting injectables. The paper is very well written and structured in a very clean way. All data is nicely presented and discussed. In principle, I like the study and the conclusions.

However, in my opinion, the study in its current form is not suitable for a journal like Nature Communications as no experimental results or prospective designs are included. Parts of this are circumvented by using an external validation set. Still, no design ideas or conclusions are taken from the results. For example, analysis from SHAP might be discussed or used to improve the properties of a compound and not just visualize the importance of a descriptor.

At the current stage, it is a solely retrospective computational analysis.

I recommend to either extend the study by experimental prospective analysis or submit to a more theoretical journal.

REVIEWER REPORT:

Reviewer #1 (Remarks to the Author):

Machine Learning Models to Accelerate the Design of Polymeric Long-Acting Injectables

Summary: The authors present their work on training machine learning (ML) models to predict the efficacy of long-acting injectables. The ML models are trained

Methodology:

Page 5, line 119. “from previously published studies”. Please add the references to the studies the first time they are mentioned. Was this data generated by the same research group? I would also appreciate a summary of what they contain.

RESPONSE: We have added the reference to the relevant papers in the manuscript as suggested. This data was collected by a variety of different research groups around the world (this is also now highlighted in the manuscript).

What happens if the models are trained on the current external validation dataset and evaluated on the current training data?

RESPONSE: As suggested, we have performed these tests and the results were found to be equivalent to the previous. The results are shown in the table below. However, based on the additional feedback, we do not have a fixed test set anymore but do a full nested CV as requested. As such the table below is no longer relevant to the study and we do not include it in the manuscript.

(A) Table summarizing the results of the leave-one-group-out cross validation from the original submission											
	RF	NGBoost	XGBoost	LightGBM	DT	PLS	k-NN	NN	MLR (Lasso)	SVR	MLR
MAE (training)	0.164	0.170	0.181	0.167	0.182	0.311	0.21	0.218	0.33	0.234	0.359
MAE (validation)	0.164	0.172	0.186	0.191	0.204	0.21	0.226	0.229	0.252	0.301	0.383
σ M	0.004	0.004	0.005	0.005	0.005	0.005	0.005	0.006	0.005	0.009	0.010

(B) Table summarizing the results of the suggested tests (i.e., previous train and test datasets swapped)											
	RF	NGBoost	XGBoost	LightGBM	DT	PLS	k-NN	NN	MLR (Lasso)	SVR	MLR
MAE (training)	0.152	0.149	0.153	0.157	0.174	0.223	0.188	0.188	0.258	0.184	0.225
MAE (validation)	0.199	0.179	0.188	0.193	0.225	0.259	0.256	0.24	0.299	0.219	0.259
σ M	0.003	0.002	0.003	0.003	0.003	0.003	0.003	0.004	0.003	0.003	0.003

The majority of the variables used are properties of the molecule that do not change (I believe only release and time does). Since the predictions are made over time, I feel the authors should explain which variables are constant and which ones aren't.

RESPONSE: This is correct, some additional input features have since been added based on the other feedback received, however this statement remains true. Only values of time vary for a given drug release profile prediction. A better description of this has been added to the input feature selection subsection of the results and discussion section.

Quality of the figures in the supplement is poor. Please update the supplement with high quality figures so that they are readable.

RESPONSE: We have updated the supplementary information document to reflect all of the changes in the current manuscript and we have improved the quality of the figures now included.

Was any of the drugs in the external dataset present in the training set? This is not mentioned and should be clarified.

RESPONSE: Yes, there were some individual drugs that were included in both the previous training and external validation datasets. There were also some polymers that were include in both the previous training and external validation datasets. However, there were no drug-polymer combinations repeated between the previous training and external validation dataset. To train the models, data is grouped by drug-polymer combination (i.e., to reflect the composition of each LAI), as this would best allow the model to be cross validated against LAI data splits. This remains true in the updated version of the manuscript where we implement the suggested nested cross-validation approach. This has also been clarified in the updated manuscript.

Can the authors provide a tsv file instead of the excel files (same matrix employed to train the ML models)? Ideally, it should be released in zenodo to be used as a benchmark for future studies. The GitHub repo should be restructured so that there are directories (Python modules) where all the models are located.

RESPONSE: We have provided a tsv file in the GitHub repo and restructured the directories in the repo. Upon acceptance, we will also release the datafile in zenodo.

Page 7 line 62. “As well as” shouldn't be preceded by a comma.

RESPONSE: All of the commas preceding “as well as” have been removed in the updated version.

The validation proposed by the authors is strong but it could be improved if the authors conduct a nested-CV approach and report the values of the test dataset as well. Then the best hyperparameters would be tested in the external dataset (Figure 1). It would be important to show that the most optimal hyperparameters match what has been reported now with a single CV.

RESPONSE: Based on this feedback we have implemented a nested cross-validation approach to train the models (we have also included some additional features as per additional feedback), the combination of both has significantly improved the accuracy of the models versus the previous leave one group out cross-validation approach with less features. The manuscript (and GitHub repo) has been updated to describe this new training approach and results.

Figure 1 bar for MLR goes out of the plot.

RESPONSE: This has been corrected, and the figure updated with the nested cross-validation data.

Given the small amount of data points, it is not surprising that NN do not perform well here. Also, given that most of the data points contain variables that are properties of the chemical, tree-based models (using thresholds) are likely to be the best performing models. This should be discussed in the conclusion.

RESPONSE: This has been added to the conclusions.

Also, which other variables could be added and why they couldn't be added in the current study.

RESPONSE: We had initially considered including initial drug release values (up to one day) as input features in these models to potentially improve performance. These were not included in the previous versions of the models, as there were no consistent drug release timepoints reported in the literature data that we collected. However, we have since imputed the missing values and included some initial drug release timepoints, which (combined with the nested cross-validation approach) has significantly improved the performance of the models.

Do drug_TPSA and Durg_NHA values come from Rdkit predictions or were collected from studies? Maybe the predictions are wrong and that's why they are irrelevant.

RESPONSE: Yes, both values are generated by Rdkit, and we agree with the reviewer that these predictions may be wrong and therefore irrelevant for that reason. This comment is now mentioned in the result and discussion section.

Conclusion

Why is it hard to collect more data points than the current ones? In other words, why are there only a few drugs that have been used for training the ML models and not over 100 for instance?

RESPONSE: Data quality is the major issue here, rather than data quantity. As mentioned above in the previous response, we encountered a lot of “missing data” when collecting data from the literature. The field seems to be limited by a lack of standardization and consistency in reporting results. In our case this

led to many potential manuscripts being discounted from inclusion in our dataset as they were missing some key feature of the drug delivery systems. For example, many manuscripts do not disclose important properties of the LAI systems, including polymer properties (such as molecular weight, lactic acid to glycolic acid ratio, drug loading levels, particle size measurements, etc). Please see the embedded table below as an example of this (i.e., a non-exhaustive snapshot of missing LAI data for a Web of Science literature search). Unlike with the aforementioned initial drug release timepoints, it is not easy to impute these properties of the polymers. We are actually in the process of writing a perspective article on this issue. This table will be included in that article, and not published in the current manuscript.

Drug	Polymer Characteristics		Initial Polymer-Drug Ratio (w/w)	Formulation Characteristics		
	Molecular Weight (kDa)	LA:GA		Size (μm)	EE (%)	DL (%)
Lidocaine	N/A	50:50	22:1	14.4	N/A	4
Doxycycline	60	50:50	1.7:1	200 \pm 1.52	24.7 \pm 0.53	N/A
Etanidazole	40-75	50:50	N/A	422.3 \pm 174.5	54.83 \pm 0.30	5
Celecoxib	N/A	85:15	N/A	1.11 \pm 0.08	51.48 \pm 0.42	14.93 \pm 0.12
Doxorubicin HCl	10	50:50	7.5:1	3.6 \pm 0.4	74.9 \pm 3.7	N/A
Simvastatin	50-75	85:15	11.5:1	3.963	96.74 \pm 1.46	N/A
Budesonide	75	50:50	10:1	16.67 \pm 1.24	74.28 \pm 0.83	N/A
Ciprofloxacin	31	50:50	22:1	5.7	8.91 \pm 2.90	0.40 \pm 0.13
Dexamethasone acetate	19	75:25	N/A	1	N/A	2.5
Leuprolide acetate	N/A	50:50	4:1	N/A	97	20

Reviewer #3 (Remarks to the Author):

The authors present different machine learning approaches for the design of long-acting injectables. The paper is very well written and structured in a very clean way. All data is nicely presented and discussed. In principle, I like the study and the conclusions.

However, in my opinion, the study in its current form is not suitable for a journal like Nature Communications as no experimental results or prospective designs are included. Parts of this are circumvented by using an external validation set. Still, no design ideas or conclusions are taken from the results. For example, analysis from SHAP might be discussed or used to improve the properties of a compound and not just visualize the importance of an descriptor.

At the current stage, it is a solely retrospective computational analysis.

I recommend to either extend the study by experimental prospective analysis or submit to a more theoretical journal.

RESPONSE: In the revised version of the manuscript, we have conducted some further model interpretation steps. Specifically, we have used the SHAP analysis to derive some key design criteria for “fast” and “slow” release PLGA system. Based on these design criteria, we have prepared and characterized (experimentally) two new LAI systems for drugs that were not previously included in our dataset. Finally, we have measured the *in vitro* drug release of these new systems and compared the

experimental release with the predicted release profiles generated by the trained model. Overall, these experimental and predicted drug release profiles were in good agreement with each other, and we found that the design criteria (which we derived from the SHAP analysis) provided excellent experimental parameters to design “fast” and “slow” release PLGA systems for new drugs.

Additionally, based on the feedback of Reviewer 1 we have re-trained the previous models via the implementation of a nested cross-validation approach and included some additional input feature for the LAI system. The combination of both has significantly improved the accuracy of the models versus the previous leave one group out cross-validation approach (which had less input features). Thus, the SHAP analysis and design criteria that we have developed are based on these updated models.

REVIEWER COMMENTS

Reviewer #1 (Remarks to the Author):

The authors have addressed all my comments. I hope that the data file will be uploaded to zenodo.

Reviewer #3 (Remarks to the Author):

Thanks a lot for the revised manuscript. The manuscript has greatly improved and the extended studies add a lot of value to the new version.

The prospective example shows the potential of the method and a possibility how to use interpretations from SHAP to design new system.

The study is very interesting and should be published in Nature Communications.

Reviewer #1 (Remarks to the Author):

The authors have addressed all my comments. I hope that the data file will be uploaded to zenodo.

Response: We thank the reviewer for their feedback. The dataset and code have been uploaded to zenodo. The manuscript has been edited to highlight this:

Data Availability

The dataset and results that support the findings of this study are available on zenodo (<https://doi.org/10.5281/zenodo.7309020>) and ChemRxiv (<https://doi.org/10.26434/chemrxiv-2021-mxxw-v2>).

Code Availability

The code that supports the findings of this study are available at the Aspuru-Guzik Group's GitHub page (<https://github.com/aspuru-guzik-group/long-acting-injectables>) and on zenodo (<https://doi.org/10.5281/zenodo.7309141>).⁶⁰

Reviewer #3 (Remarks to the Author):

Thanks a lot for the revised manuscript. The manuscript has greatly improved and the extended studies add a lot of value to the new version. The prospective example shows the potential of the method and a possibility how to use interpretations from SHAP to design new system. The study is very interesting and should be published in Nature Communications.

Response: We thank the reviewer for their feedback.